# Gout and Diet: A Comprehensive Review of Mechanisms and Management

**DOI:** 10.3390/nu14173525

**Published:** 2022-08-26

**Authors:** Yingling Zhang, Simin Chen, Man Yuan, Yu Xu, Hongxi Xu

**Affiliations:** 1School of Pharmacy, Shanghai University of Traditional Chinese Medicine, Shanghai 201203, China; 2Engineering Research Center of Shanghai College for TCM New Drug Discovery, Shanghai 201203, China; 3Shuguang Hospital, Shanghai University of Traditional Chinese Medicine, Shanghai 201203, China

**Keywords:** gout, uric acid, purine, systemic pathways, nutritional factors, dietary intervention

## Abstract

Gout is well known as an inflammatory rheumatic disease presenting with arthritis and abnormal metabolism of uric acid. The recognition of diet-induced systemic metabolic pathways have provided new mechanistic insights and potential interventions on gout progression. However, the dietary recommendations for gouty patients generally focus on food categories, with few simultaneous considerations of nutritional factors and systemic metabolism. It is worthwhile to comprehensively review the mechanistic findings and potential interventions of diet-related nutrients against the development of gout, including purine metabolism, urate deposition, and gouty inflammation. Although piecemeal modifications of various nutrients often provide incomplete dietary recommendations, understanding the role of nutritional factors in gouty development can help patients choose their healthy diet based on personal preference and disease course. The combination of dietary management and medication may potentially achieve enhanced treatment effects, especially for severe patients. Therefore, the role of dietary and nutritional factors in the development of gout is systematically reviewed to propose dietary modification strategies for gout management by: (1) reducing nutritional risk factors against metabolic syndrome; (2) supplementing with beneficial nutrients to affect uric acid metabolism and gouty inflammation; and (3) considering nutritional modification combined with medication supplementation to decrease the frequency of gout flares.

## 1. Introduction

Throughout history, gouty disease has always been strongly associated with abundant foods and immoderate alcohol intake. Gout is even known as the “king’s disease” and symbolized social status in ancient times, as only the upper class could afford to consume wine and meats [1]. Nevertheless, gout currently has been well established as a global health problem and has gained attention due to its increasing incidence rate, multiple metabolic comorbidities, and high premature mortality [2]. Gout is well known as a phlogistic arthritis that is associated with hyperuricemia and elevation of urate in tissues. The increased urate causes the generation of monosodium urate (MSU) crystals, and MSU crystal deposition in and around the first metatarsophalangeal joint, knee, and fingers represents a clinical sign of gout [3]. The clinical symptoms of gout develop in several stages, including asymptomatic hyperuricemia, MSU crystal formation, intermittent gout and chronic gout [4]. Effective gout management mainly relies on the use of therapeutic strategies to control uric acid levels or achieve crystal dissolution. While current clinical principles based on medicinal management for gout have been well implemented [5], dietary modification and lifestyle changes have also been recommended for gout patients, since a suboptimal diet and obesity/diabetes-diseases of affluence contribute significantly to the risk of developing gout [5,6,7], increasing the burden of medical expenses. At present, dietary recommendations have been updated worldwide, and nutritional science for the management of gout has advanced dramatically [8,9]. Historical dietary recommendations for gouty patients tend to emphasize the concepts of “high-” and “low-” levels of the same nutrient, but the role of nutritional elements in gout is difficult to classify by a single beneficent or harmful criterion because problems of systemic metabolism arise when the balance between nutrient intake and consumption is disturbed. This means that although dietary management is considered to be an essential aspect of gout therapeutic strategies [2], a potential dietary mechanism in gout development is out of date or incomprehensive, and a systemic overview of dietary and nutritional factors of gout is needed for well-designed dietary management based on research practice. Moreover, studies have shown that dietary factors mainly focus on food classification, while diet-induced systemic metabolism is rarely mentioned in the progression of gout. It is worthwhile to comprehensively review the mechanistic findings of diets aimed at purine metabolism, urate deposition and gouty inflammation. Our acknowledgment of gout-related nutritional factors can provide a theoretical basis for on-target and comprehensive dietary guidelines for gout patients with different complications or at different stages.

## 2. Role of Dietary Consumption in the Progression of Gouty Diseases

### 2.1. Uric Acid Disturbance: Emphasis on Purines

#### 2.1.1. Formation of Uric Acid

Purines are the basic components of nucleotides needed for building DNA and RNA within mammalian cells, and purine nucleotides, such as guano-sine-5′-triphosphate and adenosine triphosphate (ATP), are essential for regulating energy metabolism and intracellular functions, respectively. Disorders of purine metabolism are associated with considerable variations in the concentration of serum uric acid (SUA) because uric acid is the ultimate product of purine catabolism in humans. Purine source analyses show that nearly two-thirds of purines in the body are endogenous, and the remaining purines that enter the body via foods are known as exogenous purines [10]. As a direct source of exogenous nucleotides and uric acid, dietary purines are vital for maintaining the balance of purine metabolism in mammalian cells by a coordinated process of de novo biosynthesis, the salvage pathway, and purine inter-conversion or degradation. As shown in Figure 1, these exogenous purines can be dephosphorylated into nucleosides in the body as part of the digestive course, accompanied by the oxidative release of free bases [11]. Subsequently, the degraded bases can be recycled into nucleotides in the tissues via the purine salvage pathway or wholly degraded into uric acid mainly within the liver or the small intestine [11,12]. Adenosine monophosphate (AMP) and guanine monophosphate (GMP) are usually the predominant forms of purine nucleosides derived from foods. The enzymatic degradation process uses deaminase and GMP reductase to convert AMP and GMP, respectively, to inosine monophosphate (IMP) [13]. AMP/GMP can also be dephosphorylated to generate adenosine/guanosine in a process catalyzed by nucleosidase. Both IMP and adenosine are then processed into inosine. The transformation of inosine to hypoxanthine is catalyzed by purine nucleoside phosphorylase, and eventually hypoxanthine utilizes the dual oxidation of xanthine oxidase (XO) to produce uric acid. Additionally, the conversion of guanosine into uric acid can be catalyzed by guanine deaminase following guanine formation by nucleotidase [13]. When purine overload in the body empowers the body’s ability to manage it, excessive uric acid can accumulate in the bloodstream. This condition presenting with an elevated SUA concentration is known as hyperuricemia, and gout induced by hyperuricemia is deemed to be the metabolic disease linked to purines. All meats and edible plants contain purines, and some foods contain higher concentrations. Thus, overindulgent intake of a high-purine diet, including seafoods and animal offal, can trigger the excessive accumulation of purine metabolites, giving rise to the excessive accumulation of uric acid in the body [5]. In addition, some purine-free drinks can accelerate the promotion of purine degradation; for example, alcohol intake consumes large amounts of ATP to produce AMP in the liver, leading to the rapid occurrence of increased SUA levels [14]. The consumption of yeast-rich foods, such as bread and yeast drinks, can lead to a high colonization of *Saccharomyces cerevisiae* in the gut [15], which can gradually elevate the secretion of uric acid in the host.

#### 2.1.2. The Excretion of Uric Acid

Under normal conditions, nearly 90% of uric acid is reabsorbed into the human body, and the remaining uric acid is excreted in the feces and urine [16]. When purines exceed the limit value for normal production and catabolism, the synthesis and excretion of uric acid are out of balance, and the circulating uric acid level is elevated. The kidneys are responsible for eliminating approximately two-thirds of circulating uric acid, with the remaining one-third excreted by the intestine and gut microbiota [12]. Diets could affect uric acid excretion by regulating the excretory function of the kidney and intestine. Because foods can come into contact with the intestinal tract and regulate intestinal homeostasis [17], dietary factors are involved in intestinal urate handling mechanisms. Endogenous uric acid from the bloodstream or as a constituent of saliva, bile, or peptic juices transfer from the enterocyte cytoplasm into the intestine tissues. In this process, enterocyte urate transporters are critical for maintaining urate homeostasis in the intestine, and an enterocyte-specific deficiency of these transporters, including ATP-binding cassette transporter G2 (ABCG2/BCRP) and NPT5 (SLC17A4), can impair the enterocyte urate transport process, which is also affected by exogenous metabolism, such as dietary fat and sugar [18]. Similarly, a study of 8709 participants suggested that high simple sugar exposure was found to interfere with the ability of SLC2A9 (encoding GLUT9) to mediate renal uric acid excretion without additive genotype-specific interaction [19]. After entering the intestinal tissue, uric acid can be degraded into nitrogen or CO_2_ by the uricase activity found in the gut microbiota. The associations between diet-induced gut microbiota reconstruction and the progression of hyperuricemia/gout have been highlighted in recent research, as evidenced by the fact that long-term adherence to the typical Western diet caused an obvious reduction in the diversity of the gut microbiota, particularly those that degrade uric acid and produce metabolites known to benefit uric acid excretion [20]. For example, as a microbiota-derived metabolite, the short-chain fatty acid butyric acid was thought to promote intestinal uric acid excretion [21]; however, a fat-rich diet reduces the abundance of beneficial bacteria that produce short-chain fatty acids [20]. In addition, the diet is mainly designed to provide calories for energy expenditure, and the related energy metabolism and metabolites produced as a result of dietary modification can affect uric acid excretion. By way of illustration, a ketogenic diet converts energy metabolism substrates from sugars to fats with the production of large ketone bodies such as acetyl acetate and β-hydroxybutyrate (BHB) [22], which induce fluid acidification and cause uric acid precipitation. The production of BHB inhibits uric acid excretion by competing for binding sites of uric acid transporters.

### 2.2. Uric Acid Disturbance: Emphasis on Purines

Monosodium urate crystals formation occurs when the urate concentrations continue to rise beyond the point required for spontaneous generation (nucleation). The increased volume of MSU deposition can aggravate the progression of symptomatic gout [23]. Diets that contribute to excessive SUA levels can induce the formation and deposition of MSU crystals, and other factor changes, such as diet-induced fluid acidification and salt deposition, also promote the growth of urate crystallization [24]. Alcohol consumption and fasting can induce elevated lactic acid levels to decrease the local pH to create an acidic condition, which might be a risk factor for MSU deposition, as the increased concentrations of calcium ions in an acidic environment aggravate the decrease in MSU crystal solubility [25].

### 2.3. Gouty Inflammation

Some observations show that in some circumstances, uric acid can show antioxidant properties in the form of urate. However, soluble urate acts as a proinflammatory stimulus to fuel the maturation and production of interleukin (IL)-1β [26], thus strongly driving acute gouty inflammation and giving rise to chronic long-term inflammatory consequences of the disease. It is known that the pathogenesis of gouty inflammation involves the cleavage of C5 and generation of C5a and C5b-9 on the surface of urate crystals [27]. Urate crystal deposition is also extensively recognized as a danger signal for the influx of innate immune cells [28]. Mechanistically, MSU crystal-induced inflammatory gouty flares are caused by the activation of pyrin domain-containing protein 3 inflammasome (NLRP3) with consequent IL-1β secretion from macrophages and neutrophils, resulting in acute inflammatory responses, intense pain and joint swelling [28]. The initiation of NLRP3-dependent IL-1β activation includes a priming signal associated with nuclear factor κB (NF-κB) and a secondary signal of caspase-1 assembly activation; thus, MSU crystals alone are insufficient to induce IL-1β secretion. Additionally, other endogenous costimulatory factors, such as myeloid-related protein-8/-14 in phagocytes, can increase MSU crystals-mediated IL-1β secretion in a TLR-4-dependent pathway [27].

Diets also have a significant impact on the systemic phenotype of the innate immune system. For example, a Western diet or meat-based patterns intersect with a low-grade inflammatory response, which permanently biases the immune system toward a proinflammatory phenotype [29], fueling gouty inflammation. In contrast, the Mediterranean diet or vegetable- and fruit-based patterns have been reported to prevent systemic inflammation and gouty flares [30]. A fiber-rich diet has been proven to rapidly resolve the urate crystal-mediated inflammatory response in a gout-like mouse model [31]. Diet intervention affecting the neutrophil inflammasome has been explored. For example, a ketogenic diet alleviates urate crystal-induced gouty flares by increasing BHB, which can block NLRP3/caspase-1-dependent IL-1β expression in neutrophils and urate crystal-activated macrophages, reducing inflammatory neutrophil recruitment [32].

## 3. Nutrient Element-Richness and Structure Determine the Role of Dietary Factors in Gout

A large amount of clinical evidence, shown in Table 1, indicated the close connection between adherence to the described dietary patterns and the risk of gout-related metabolic disorders, hyperuricemia, and metabolic syndrome. Commonly described dietary patterns, such as a high-carbohydrate diet, a high-protein diet, and a high-unsaturated fat diet, actually represent the proportional and structural collocation of dietary nutrient elements [33]. It has been shown that the beneficial dietary patterns against hyperuricemia usually contain a higher intake of vitamins, fiber, and unsaturated fatty acids and are often supplemented with appropriate amounts of minerals and high-quality protein, promoting a health state in which systemic metabolism is prone to disease improvement [34,35]. These observations can be explained by the fact that important nutrients in foods can be considered determinants of dietary factors in gouty development. For example, the Western diet is characterized by a high level of sugar, while fiber richness in the Dietary Approaches to Stop Hypertension (DASH) diet can increase satiety to reduce sugar intake from other high-energy foods and play a vital role in decreasing gout incidence [34]. Therefore, the influence of dietary factors on gouty disease is the effect of nutrient element-richness and structures on systemic metabolism. Understanding the potential mechanisms of nutrients, as shown in Figure 2, in gout development can facilitate an understanding of the overall nutritional balance needed for the prevention or treatment of gout.

### 3.1. Energy-Type Nutrition Overload Can Induce Hyperuricemia and Inflammation

#### 3.1.1. High Fat

Dietary fat is primarily metabolized into triglycerides within intestines and packaged as chylomicrons for delivery to peripheral tissues, where adipocytes further transfer triglycerides into free fatty acids (FFAs) for energy uptake and storage [67]. A small amount of FFAs can be absorbed by the liver tissue, together with lipids remaining in the chylomicron [67]. High-fat foods can evoke the pleasure of eating and promote the individual’s desire to consume more energy-dense diets [68], which culminates in the overproduction of FFAs. Free fatty acids-mediated metabolic events initiate acute onset of gouty disease in the presence of MSU crystals deposited in the joint. The interaction of FFAs with TLR2 synergized with MSU crystals leads to the release of IL-1β induced by ASC/caspase 1 [69].

High fat consumption can cause excessive accumulation of triglycerides, inducing increased fat mass and obesity. It has been reported that overweight/obesity was connected with 60% of hyperuricemia cases in a clinical trial of 14,624 adults [70], possibly due to lipid metabolic disorder promoting purine metabolism by elevating XO activity [71]. Hypertrophy and hyperplasia of adipocytes are associated with higher oxygen consumption, which evokes hypoxic damage in other tissues, resulting in the chronic inflammation of obesity [72]. Nonalcoholic fatty liver disease and nonalcoholic fatty pancreatic disease subsequently occur when lipid overload occurs in the liver and pancreatic tissue, causing metabolic dysfunction in both and affecting acid-base balance. Metabolic acidosis further promotes hypercalciuria, low urine pH and hypocitraturia, predisposing patients to MSU crystal deposition and calcium renal stone formation [73]. In response to excessive FFAs circulation, insulin secreted from pancreatic β-cells upregulates the expression of renal urate transporters, including GLUT9 and URAT1, and decreases ABCG2 levels, promoting high SUA levels [18]. Uric acid conversely induces lipid accumulation and insulin resistance (IR), thereby forming a vicious cycle of uric acid and insulin [74].

Prior to the onset of IR and obesity, high fat intake has been found to upregulate the expression of reactive oxygen species (ROS) in adipose tissue and liver, along with the metabolic disturbance of adipocytes and the dysregulation of adipokine release [75], promoting or aggravating MSU-mediated NF-κB-dependent inflammation. For example, leptin levels secreted from adipose tissues were elevated in patients with gouty inflammation, and leptin can facilitate MSU-induced acute gout-related proinflammatory cytokine production in macrophages and synoviocytes [76]. The key adipocyte-derived chemokines McP-1 and LTB4 recruit proinflammatory macrophages to induce inflammation amplification, thus aggravating gouty inflammation [77]. Furthermore, a high-fat diet changes gut microbiota composition, leading to reduced microbiota diversity and an increased ratio of Firmicutes to Bacteroidetes and to the reduction of microbiota-derived beneficial metabolites such as butyric acid, which further aggravates gouty arthritis [20].

#### 3.1.2. High Sugar

Sugars are the most abundant macromolecules in nature and can be classified according to their structure into monosaccharides, complex carbohydrates, and glycoconjugates [78]. They are the primary carbon source for ATP production and cellular biosynthesis. Sugars from diets can be absorbed as glucose, galactose or fructose in the liver portal circulation. The liver and gut normally process galactose and fructose into lactate, glucose and organic acids through gluconeogenesis, glycogenolysis, aerobic oxidation and other pathways [78]. High sugar consumption might initiate metabolic disease processes accompanied with hyperglycemia, IR, and fat accumulation. Moreover, high sugar intake in obese patients increases serum urate and decreases the percent of uric acid to creatinine clearance, indicating a close association between hyperuricemia and a high sugar diet [79]. Sugar-sweetened beverages containing high-fructose corn syrup and sucrose or almost equal amounts of fructose and glucose, which account for approximately one-third of added sugar consumption in the diets of American adults [80], have been thought to be closely connected with a high prevalence of hyperuricemia in Western countries [81]. Long-term high sugar consumption has been found to accelerate the accumulation of uric acid and promote MSU deposition in fly renal tubules, suggesting that a similar problem may occur in human excretory systems under dietary challenges [82]. In a follow-up study of 650 participants, the results confirmed that a high-sugar diet participates in kidney dysfunction and uric acid metabolism disorders [82].

The metabolic effects of sugar are distinct from those of starch principally because of the fructose component. Studies of dietary sugar intervention in animals and humans have demonstrated that overconsumption of fructose, but not glucose, can manifest multiple traits of metabolic syndrome [83,84,85], indicating that fructose might be responsible for high sugar-driven hyperuricemia and gout [86]. Fructose metabolism starts within the small intestinal tissue, where fructose can be absorbed by the facilitative hexose transporter GLUT5 (SLC2A5) and converted by ketohexokinase. Notably, exposure to high fructose increases the intestinal villus length to expand the surface area of intestinal cells that can absorb more nutrients from food [87], possibly aggravating high fructose-mediated metabolic disorders via overconsumption of nutrients. Excessive fructose consumption-induced gouty syndrome is related to the altered gut microbiota and its metabolites and induces inflammation and fatty acid disorders. Since high fructose intake induces the proliferation of mucus-degrading bacteria in the gut microbiota, decreased mucus glycoproteins can promote intestinal barrier damage and pathogen invasion [88].

Fructose-derived metabolites can be transferred from the intestinal tissues to the liver and systemic circulation, and the redundant fructose can also directly reach the hepatic tissue or enter the systemic circulation when the intestinal clearance capacity reaches an upper limit [89]. Fructose culminates the main rate-limiting step of glycolysis and is rapidly converted into ketohexokinase to generate fructose-1-phosphate, which is further metabolized into glyceraldehyde 3-phosphate and dihydroxyacetone phosphate. The acute fructose overload in the liver leads to ATP degradation and a decrease in ATP synthesis, both of which lead to an increase in AMP levels and stimulation of AMP deaminase activity, subsequently accelerating the formation of uric acid with consequent hyperuricemia. More notably, the costly fructose metabolism can also result in renal inflammation and fibrosis and form kidney stones due to calcium salt precipitation in high-fructose diet-fed mice [90]. Alarmingly, a normal physiological concentration of fructose in the kidney still causes a risk of defective elimination of uric acid and activation of renal inflammation by increasing the expression of intercellular adhesion molecule-1 in the serum and endothelial cells [91]. Therefore, carbohydrate restrictions, especially fructose intake, have been regarded as an efficient diet intervention for gouty patients to modulate the disease state. It has been shown that the intake of fructose-rich fruits could bring about a temporary upregulation of uric acid. However, the moderate consumption of these foods over a long time can facilitate the excretion of uric acid, which could be correlated with the alkalization of body fluids [92]. Therefore, it is always suggested that limiting fructose-rich soft drinks or reducing the consumption of high-fructose drinks rather than fruits is better for gout improvement.

#### 3.1.3. High Protein

Proteins are biomacromolecules formed by folding long chains of amino acids, and dietary protein digestion in the gastrointestinal tract can provide amino acid dipeptides and tripeptides for cellular metabolism and protein synthesis in muscle or other tissues. High protein consumption is well known for its promotion of energy expenditure and urea synthesis. In addition, high dietary protein intake can also affect uric acid homeostasis, since protein digestion can generate several amino acids, such as glutamine, glycine and threonine, to induce purine synthesis, promoting the development of hyperuricemia [93]. A cohort study with 193,676 participants further revealed that higher nondairy animal protein consumption results in a disturbance of uric acid metabolism, including a reduced level of citrate and a higher level of uric acid and acidic urine, which subsequently promotes uric acid stone formation [94]. In addition, long-term high dietary protein intake has been shown to have adverse effects on uric acid elimination due to increased intraglomerular pressure and flow in kidney tissues [95]. The severity of glomerular damage has been significantly attributed to elevated SUA levels; hence, dietary recommendations for gouty patients often suggest that excessive protein intake should be restricted to avoid placing kidney tissue under undue stress.

Studies have also shown inconsistent effects of dietary proteins from different sources [96]. Overconsumption of animal proteins is linked to an elevated prevalence of gout, whereas overconsumption of plant protein (soybeans and soy products) or dairy product intake are associated with a reduced risk [8,96]. When compared with foods rich in plant proteins, an animal protein-rich diet promotes acidic urine production and uric acid stone formation [94]. Conversely, substitution of plant-based protein for a carbohydrate-rich diet can attenuate IR and compensatory hyperinsulinemia [97], thus improving the renal clearance of urate. Therefore, choosing appropriate dietary protein sources and controlling the amount of protein intake might represent an effective intervention for the improvement of gouty diseases.

### 3.2. Adequate Consumption of Essential Nutritional Elements Leads to Beneficial Effects against Gout

#### 3.2.1. Vitamins

Vitamins are essential trace elements that act as regulators of physiological and pathological functions, such as participating in immune responses, antioxidant activities and redox reactions. It has been demonstrated that an adequate intake of vitamin supplements or consumption of vitamin-rich fruits and vegetables seems to be a valid approach for hyperuricemia and gout treatment. Vitamins such as vitamin A, vitamin E, and vitamin C show beneficial effects against oxidative stress and inflammation, as well as effectively decreasing SUA levels [98,99,100], and the same uric acid-lowering effect also appears in a vitamin D-rich diet [101]. Additionally, vitamin E is also considered a membrane stabilizer that inhibits MSU crystal-induced hemolysis [102]. Many studies performed in humans and animals have shown that vitamin C (l-ascorbic acid) consumption can affect uric acid reabsorption and excretion to reduce SUA levels [100]. Both uric acid and vitamin C can be reabsorbed in the proximal tubule via anion-exchange transport, and vitamin C overload can competitively suppress the reabsorption of uric acid [103] in the filtrate. Meanwhile, its downregulation of URAT1 activity and/or Na^+^-dependent anion cotransporter could promote uric acid excretion [100]. The uricosuric function of vitamin C also appears to directly act on the glomerulus by reducing glomerular microvascular ischemia and increasing afferent arteriole dilation, thus increasing the glomerular uric acid filtration rate [104]. Furthermore, vitamin C reduces the incidence of gout by alleviating the NF-κB/NLRP3-related inflammatory response to MSU deposition [105].

#### 3.2.2. Minerals

Minerals, including potassium, zinc, calcium, copper, iron, and selenium are micronutrients that are essential for body metabolism [73], and deficiencies or excesses of these micronutrients are potentially hazardous occurrences that might be involved in the development of gout. It is well known that dietary potassium consumption has obvious diuretic and natriuretic effects, and even a minor potassium insufficiency triggers an impairment in the kidney’s capacity to secrete sodium chloride and retain sodium [106], resulting in renal dysfunction, while long-term routine potassium replenishment aggravates thiazide diuretic-mediated elevation of uric acid [107]. A similar facilitation effect was observed when iron accumulation triggers increased saturated transferrin-mediated XO activity [108,109]. Minerals also have a crucial role in maintaining acid-base balance. This has been attributed to keeping urine electrically neutral by regulating the secretion of anions such as chloride, sulfate, and phosphate in kidney tissues [73]. For instance, urinary calcium loss is a crucial risk factor that can trigger calcium stone formation and cause a uric acid excretion disorder [73]. Normal calcium intake can decrease the potential risk of kidney stone formation and is conducive to uric acid elimination in renal tissue [110,111].

#### 3.2.3. Fibers

Dietary fibers refer to plant-derived carbohydrates that are resistant to hydrolyzation or assimilation in the upper gastrointestinal tract, and fibers trapped in the gut can increase intestinal viscosity and satiety, as well as reduce gastric emptying rate and regulate intestinal conduction [112]. Correspondingly, dietary fibers reduce the intake and absorption of high-energy food [113]. The consumption of dietary fibers can manage glucose and lipid metabolism to regulate energy balance [113,114]. More importantly, a lack of these fibers leads to a slower recovery of gut dysbiosis, and dietary fiber supplementation is able to improve the composition of the gut microbiota [115], suggesting a close connection between fiber and intestinal flora disorder in gout patients. The fiber fermentation process by gut microbiota accompanies the release of microbiota-driven metabolites (short-chain fatty acids, SCFAs) that show beneficial effects on the host’s health [116]. After dietary fiber intake, acetate is the most abundant microbiota-derived SCFAs in the blood and can quickly resolve MSU-induced inflammation by promoting caspase-dependent apoptosis of neutrophils and the excretion of the anti-inflammatory IL-10 against LPS-induced inflammation [31]. Butyrate is another SCFAs mainly produced by microbiotal fermentation of indigestible fibers, and has been shown to improve lipid accumulation in the liver and pancreas, thereby reducing uric acid metabolism abnormalities by XO activation [71]. Butyrate can also decrease the activation of NF-κB induced by LPS and the translation or transcription of IL-1β by inhibiting histone deacetylases in human peripheral blood mononuclear cells [117]. Therefore, the consumption of more fiber-rich whole grains, vegetables and fruits is beneficial for regulating gastrointestinal homeostasis, reducing the intake of unhealthy foods and reshaping the gut microbiota. Meanwhile, the salutary metabolites produced by the microbiota-induced digestion of dietary fiber can regulate the inflammatory state of gouty patients and reduce uric acid production, all of which are conducive to the management of gout.

## 4. Recommended Nutritional Management and Its Combination with Drug Therapy

The close association of specific foods with SUA levels, as shown in Table 1, illustrates that the essential impact of food on health is ascribed to the synthesized effects of the nutrients from food, specifically the dominant or beneficial nutritional factors determining the final performance. Taking dairy products as an example, the increased intake of dairy products, especially those with low fat, can reduce the incidence of gout [8]. Late season skim milk, which contains higher levels of orotic acid than early season skim milk, has a preferential impact on the excretion of uric acid [118]. Moreover, glycomacropeptide and G600 milk fat addition in skim milk can greatly and effectively relieve joint pain and the frequency of gout flares [119].

The dietary management for gouty patients is commonly self-prescribed and centers around the control of purine sources, such as reduced consumption of purine-rich foods, which theoretically attenuates uric acid production; but patients often fail to adhere to recommendations in the long term due to the limited palatability of purine-free diets [34,37]. Although piecemeal modifications of the various yet limited numbers of nutrients often provide incomplete dietary recommendations [34], attention should be given to nutrient richness and structure to avoid the burden of inappropriate dosage. For example, the plasma concentrations of vitamin C saturation ranges daily from 200 to 400 mg, implying that exceeding the recommended supplemental dose has little effect on the consequences [120]. More importantly, taking high-dose and long-term supplements of vitamin C may be associated with adverse effects, and the resulting excessive uric acid excretion could elevate the risk of kidney stones in gouty patients [121,122]. As shown in Figure 3, recommended nutritional interventions should emphasize the necessity for appropriate supplementation of plant-derived fibers and dairy-derived protein and persuade patients to avoid high-compensation consumption of refined saturated fats and carbohydrates [34]. The typical dietary patterns include a DASH and Mediterranean diet, both of which are comprised of fruits, vegetables, and low-fat dairy products with reductions in total and saturated fats. Increasing evidence supports that consuming a DASH diet can continuously attenuate SUA in hyperuricemia patients and reduce the incidence of gout in participants [34,45]. Similar SUA-lowering effects have been observed in a research investigations of the Mediterranean diet [58]. Moreover, intervention with the DASH diet combined with adequate sodium and plant-derived protein shows more beneficial functions in reducing SUA levels [33,37].

Most cases of dietary management related to uric acid reduction are reported in the nongouty population (as shown in Table 1). Dietary modification against gouty disease is commonly selected as an adjunct therapeutic strategy with medicinal drug therapy [10]; however, clinical trial data on the effects of dietary modifications combined with pharmacological interventions on gout are mostly lacking, as shown in Table 2. By way of illustration, colchicine and nonsteroidal anti-inflammatory agents (NSAIDs) have been used to treat acute gouty inflammation. Their combined use with dietary micronutrient supplementation can remedy micronutrient deficiency induced by NSAIDs and colchicine [123,124]. Allopurinol, a first-line drug used as a xanthine oxidase inhibitor against gout, requires high protein replenishment to facilitate renal clearance [125]. Therefore, our recommendation is for individuals to follow a healthy diet for prevention purposes, and for patients with mild gout, we recommend the DASH and Mediterranean diet, which focus on plant-based components. Additionally, we recommend a reduction in the consumption of high-fat foods (fast food and cream products), especially foods with trans fatty acids (such as margarine and butter), and for individuals to pay attention to the amount of nutrient supplements consumed. For patients with severe gout, dietary modification and medication should be combined, and health care providers should remind patients of food–drug interactions to achieve synergistic effects.

## 5. Conclusions and Future Perspective

Our study has shown the important role of diet in gout development and management and how dietary adjustments based on nutrient composition should be an important component of routine care for gout. Gout patients are commonly prone to choose health management by dietary modification because diet changes can instantly affect gout flares via multiple signaling pathways. Diet-induced systemic metabolic pathways, including purine, lipid, and glucose metabolism, as well as energy balance and gut microbiota changes, have provided new mechanistic insights and potential interventions for gout progression. Since foods are eventually metabolized into multiple nutrients for metabolic homeostasis in the body, dietary modification might represent an appropriate nutritional regulation for gout patients or for potential patients to effectively reduce the incidence of gout. The critical role of nutritional factors on gout development also supported the following recommended nutritional modification strategies: (1) reducing nutritional risk factors against metabolic syndrome; (2) supplementing with beneficial nutrients to affect uric acid metabolism and gouty inflammation; and (3) considering nutritional modification combined with medication supplementation to decrease the frequency of gout flares. Our consistent principle is that, in terms of diet, nutritional balance should be analyzed from the point of view of enrichments and structures of nutritional elements. Evidence supports that a low-fat, low-carb, plant-based dietary intervention is suitable for gouty patients; however, we need to pay specific attention to the golden rule of healthy dietary intake, that is, moderation. In addition, we advocate the combination of medications and dietary modification for gouty patients in therapy, and caution that they should note the impact of nutritional factors on drug pharmacokinetics and pharmacodynamics. However, many human studies focusing on the relationship of food and SUA levels often do not involve gouty patients. A limited number of clinical research studies explore food/nutrition and gout or anti-gout drugs, resulting in limited being drawn conclusions. In conclusion, the dietary mechanisms and nutritional basis provide scientific evidence for the prevention and improvement of gouty diseases, and dietary modifications based on effective regulatory mechanisms may be a promising strategy to reduce the high prevalence of gout.

## Figures and Tables

**Figure 1 nutrients-14-03525-f001:**
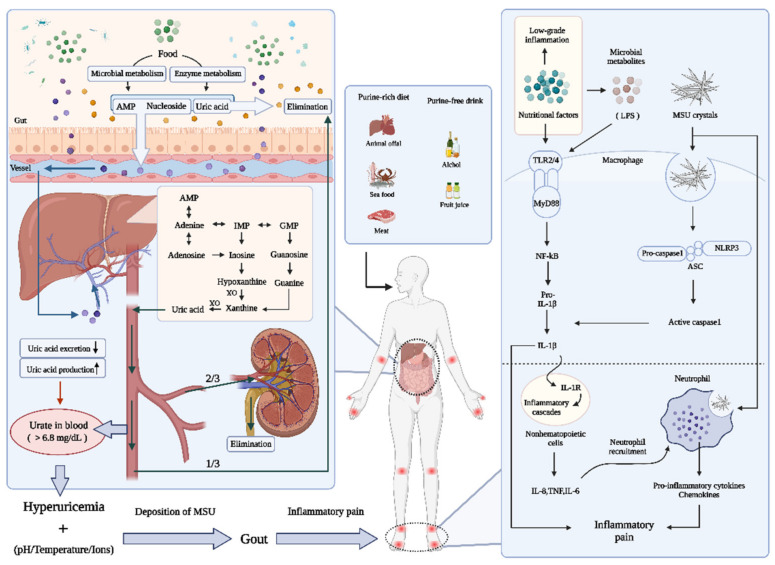
Potential mechanisms of diet-induced gout progression in humans. Diets provide abundant raw materials of purine, which is mainly metabolized in the liver, promoting uric acid production. Meanwhile, it can interfere with the intestinal environment, homeostasis, and urate transport to induce high levels of uric acid, leading to hyperuricemia and ultimately to gout. Additionally, gouty inflammation is caused by IL-1β production after the activation of NLRP3 by macrophages that ingest MSU crystals, and a second signal is required in humans by stimulating the activation of TLR signaling pathways that can be induced by diets. Moreover, neutrophil infiltration and diet-induced low-grade inflammatory states will exacerbate gouty inflammation. AMP, adenosine monophosphate. ASC, apoptosis-associated speck-like protein containing a caspase recruitment domain. GMP, guanine monophosphate. IL, interleukin. IMP, inosine monophosphate. LPS, lipopolysaccharide. MSU, monosodium urate. MyD88, myeloid differentiation factor88. NF-κB, nuclear factor kappa B. NLRP3, pyrin domain-containing protein 3. TLR, toll-like receptor. TNF, tumor necrosis factor. XO, xanthine oxidase.

**Figure 2 nutrients-14-03525-f002:**
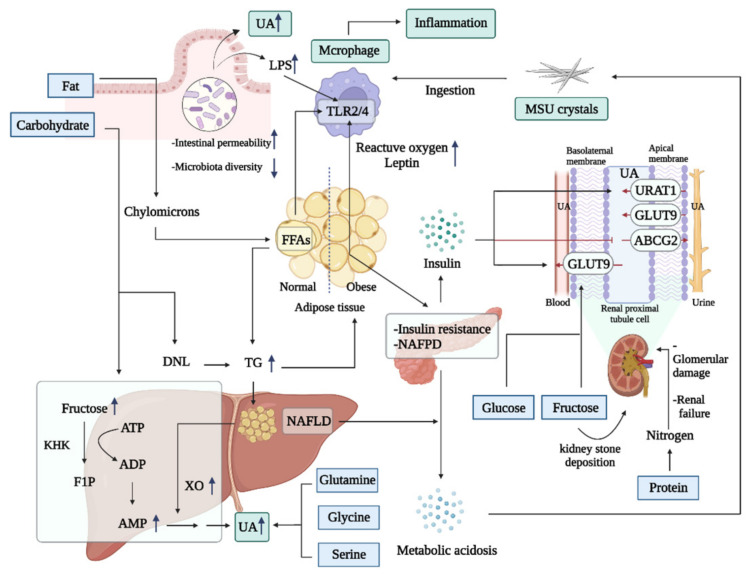
Nutrition-induced systemic metabolism involved in gouty disease. Metabolites of fat, carbohydrate and protein and the resulting metabolic diseases promote the development of gout, including changing intestinal flora, accelerating purine metabolism, promoting MSU deposition, activating macrophages, and inhibiting uric acid excretion. ADP, adenosine diphosphate. AMP, adenosine monophosphate. ATP, adenosine triphosphate. FFAs, free fatty acids. F6P, fructose 6 phosphate. KHK, ketohexokinase. LPS, lipopolysaccharide. MSU, monosodium urate. NAFLD, nonalcoholic fatty liver disease. NAFPD, nonalcoholic fatty pancreas disease. TG, triglyceride. TLR, toll-like receptor. UA, uric acid. XO, xanthine oxidase.

**Figure 3 nutrients-14-03525-f003:**
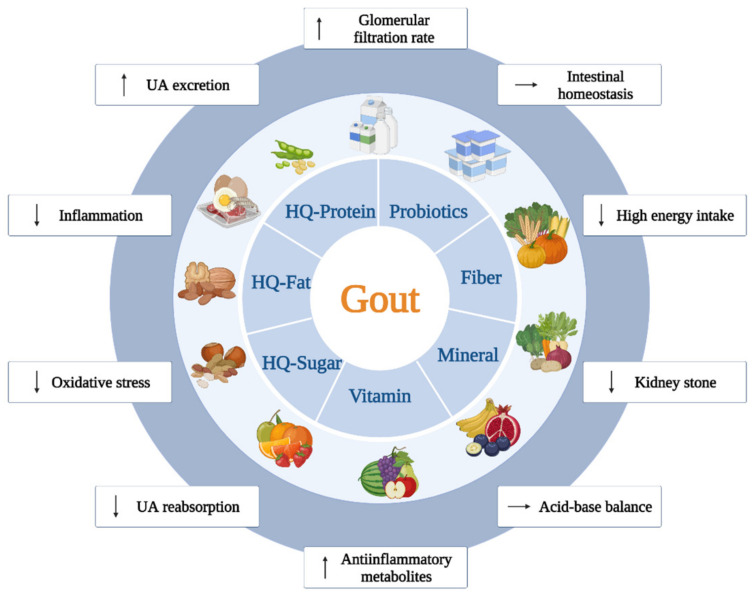
Recommended food-derived nutritional interventions with anti-gouty mechanisms. Dietary management recommendations for gout patients include appropriate intake of fiber, minerals, and vitamins, as well as the selection of high-quality sugars, fats, and proteins, which are usually of plant origin. In addition, the consumption of products containing probiotics helps regulate intestinal homeostasis in patients with gout. ↑—increase; ↓—decrease; →—maintain; UA—uric acid; HQ—high-quality.

**Table 1 nutrients-14-03525-t001:** Dietary intervention associated with the improvement of serum uric acid and indicators of metabolic syndrome.

Intervention Group	Control Group	Participants	Period	Major Findings
Low-carbohydrate (≤20 g/day) and high-fat diet [36]	Habitual diet (carbohydrate ≤ 20 g/day)	30 heathy persons (ages ≥ 18 years)	3 weeks	UAM: urate significantly ↑ in the LCHF group MS in the LCHF group: •apolipoprotein B, TC, HDL-C significantly ↑ •FFA and urea significantly ↑ •mean plasma LDL-C ↑
DASH diet with low, medium, and high sodium levels [37]	The average American diet	103 subjects (average age of 51.5 years) with pre- or stage 1 hypertension	30 days	UAM: •mean SUA ↓ in the DASH diet group vs. the control group •SUA ↓ in medium and high sodium intake when aggregated across both diets
Fruit-rich and soybean products diet (Group 1) [38]	Standard diet for hyperuricemia (Group 2)	187 Chinese adults (ages 20 to 59 years) with asymptomatic hyperuricemia	3 months	UAM: SUA ↓ in the Group 1 and Group 2 vs. baseline MS: •HDL-C significantly ↑ in the Group 1 vs. the baseline •BMI, TC and TG significantly ↓ in the Group 2 vs. the baseline
Low-salt diet followed by a high-salt diet [39]	/	90 subjects with similar dietary habits (ages 18 to 65 years)	17 days	UAM: •PUA significantly ↑ in the low-salt diet group and PUA significantly ↓ in the high-salt diet group vs. baseline •24 h UUA significantly ↓ in the low-salt diet group and the high-salt diet group vs. baseline
2 apples/day for 8 weeks, and then after a 4-weeks Washout period, consumed 500 mL of control beverage daily for a further 8 weeks (Group 1), or received the intervention foods in the reverse order (Group 2) [40]	/	40 healthy and mildly hypercholesterolemic Volunteers (ages 29 to 65 years)	20 weeks	UAM: SUA ↑ in the Group 1 vs. the Group 2 MS: TC, LDL-C, TG and ICAM-1 significantly ↓ in the Group 1 vs. the Group 2
Regular cola (SSSD); Diet cola; Isocaloric semiskimmed milk; Water [41]	/	47 overweight and obese adults (ages 20 to 50 years)	6 months	UAM: PUA significantly ↑ in the SSSD group vs. other groups MS: •VAT significantly ↑ in the SSSD group vs. other beverages, and in liver fat of more than two-fold •Plasma TG ↑ in the SSSD group vs. the milk, the diet cola and the water group
High-carbohydrate diet (CARB); High-protein diet (PROT); High-unsaturated fat diet (UNSAT) [33]	/	163 subjects (ages ≥ 30 years)	6 weeks	UAM: •SUA ↓ in PROT group vs. baseline •SUA significantly ↓ in PROT group vs. the CARB and UNSAT group
Pakistani almonds (PA); American almonds (AA) [42]	No intervention	150 patients with coronary artery disease (ages 55 to 63 years)	12 weeks	UAM: •SUA ↓ in the PA group and the AA group at week 6 and week 12 vs. the NI group
High-carbohydrate and high/low-glycemic index diet (CG/Cg); low-carbohydrate and high/low-glycemic index diet (cG/cg) [43]	/	163 overweight or obese adults without cardiovascular disease (ages ≥ 30 years)	5 weeks	UAM: •PUA ↓ in the Cg group and PUA ↑ in the cG group vs. baseline •PUA ↓ in the Cg group vs. the CG group •PUA ↓ in the cg group vs. the cG group •PUA ↑ in the cG group vs. the CG group
Yogurt with 300 g/day of probiotic [44]	Regular yogurt	44 metabolic syndrome patients (ages 20 to 65 years)	8 weeks	UAM: •SUA ↓in the probiotic yogurt group •significantly changes in UA level MS in the probiotic yogurt group: •MDA and oxidized LDL ↓ •TAC ↑
Fruit and vegetable (FV)-rich diet; DASH diet [45]	Typical American diet	459 subjects with blood pressure (<160 mmHg, 80–95 mmHg) (ages ≥ 30 years)	8 weeks	UAM: •SUA ↓ in the FV group and SUA ↓ in the DASH group •effects increased in DASH group with increasing baseline SU levels
100% orange juice; caffeine-free cola [46]	/	26 healthy adults have a habitual three-meals-per-day structure (ages 20 to 45 years)	2 weeks	UAM: SUA significantly ↓ and UUA significant ↑ in the orange juice group vs. baseline MS: daylong glycemia and glucose variability significantly ↑, 24 h insulin secretion and serum potassium levels significantly ↓ in the cola group vs. orange juice group
High-resistant starch with low-protein flour staple (Group 1) [47]	Protein-restriction diet	75 patients with early type 2 diabetic nephropathy (ages 18 to 80 years)	12 weeks	UAM: SUA ↓ in the Group 1 MS: fasting BG, HbA1c, TC and TG significantly ↓ in the Group 1; serum superoxide dismutase level b2-microglobulin ↑ in the Group 1
Sugar-sweetened soda or reduced-fat milk [48]	/	30 overweight or obese subjects (males, ages 13 to 18 years)	Not specified	UAM after the milk intake phase: UA significantly ↓ MS after the milk intake phase: systolic blood pressure significantly ↓ after the milk intake phase
DASH diet followed by self-directed grocery purchases (DDG) or the reverse order (SDG) [49]	/	43 gouty participants without taking urate lowering therapy (ages ≥ 18 years)	8 weeks	UAM: •SUA ↓ in the DDG group during Period 1 •SUA ↓ in the SDG group and SUA ↓ in the DDG group after crossover (Period 2) MS: total spot urine sodium excretion ↓ in the DDG group
Standard metabolic diet (beef, fish, or chicken) [50]	/	15 healthy subjects (ages 18 to 70 years)	Not specified	UAM: •SUA significantly ↑ for each diet phase, and beef was associated with lower SUA than chicken or fish •fish was associated with significant UUA ↑ than beef or chicken •calcium oxalate significantly ↑ in the beef diet phase vs. the chicken diet phase
3 servings of 100% naturally sweetened orange juice (OJ)/day [51]	3 servings of sucrose-sweetened beverages (sucrose-SB)/day	20 healthy and overweight women (ages 25 to 40 years)	2 weeks	UAM: PUA significantly ↑ in the sucrose-SB group, and PUA ↓ in the OJ group vs. AUC of baseline MS: •BW significantly ↑ in the sucrose-SB group vs. baseline •BW ↑ in the sucrose-SB group vs. OJ group •Matsuda insulin sensitivity index ↓ in both group
High-fructose corn syrup (HFCS): 0% (aspartame sweetened), 10%, 17.5%, 25% Ereq-HFCS [52]	/	187 participants (ages 18 to 40 years)	2 weeks	UAM: 24-h mean PUA significantly ↓ in 10%, 17.5% and 25% HFCS group vs. the 0% group MS: postprandial TG and fasting LDL-C significantly ↑ in 10%, 17.5% and 25% HFCS group vs. the 0% group
Tomatoes [53]	/	35 Caucasian women (ages 18 to 25 years)	4 weeks	UAM: PUA ↓ vs. baseline MS: mean BW, fasting BG, TG, C ↓ vs. baseline
High-calcium fat-free milk session and followed by consumption of low-Ca control session (HC group) or the reverse order (LC group) [54]	/	14 type 2 diabetes subjects with habitual low calcium intake (ages 20 to 59 years)	32 weeks	UA: SUA ↓ in the HC group and SUA significantly ↑ in the LC group MS: •25-hydroxyvitamin D significantly ↑, fructosamine and parathormone significantly ↓ in the HC group • 25-hydroxy-vitamin D significantly ↑ in the HC group vs. the LC group • Hb1Ac significantly ↑ and HOMA2-%B significantly ↓ in the LC group
500 mL orange beverage (OB)/day [55]	Not consume OB	30 healthy volunteers (average age of 33.9 years)	2 weeks	UAM: PUA significantly ↓ in the OB intervention phase vs. both of baseline and washout phase MS: • ORAC ↑ while CAT, TBARS and *C*-reactive protein ↓ in the OB intervene phase vs. baseline •CAT, TBARS and oxidized LDL ↓ after the wash out phase vs. baseline
High-fructose or high-glucose diet [56]	/	32 healthy but centrally overweight men (ages 18 to 50 years)	10 weeks	UAM: SUA ↑ in the fructose group SUA ↓ in the glucose group MS: •the risk of insulin resistance ↑ in the fructose diet group vs. the glucose diet •BG, TAG and biochemical assays of liver function ↑ in both group
Diet rich in whole grain (WG) products for 3 weeks followed by red meat (RM), or the reverse order [57]	/	20 healthy adults (ages 20 to 60 years)	10 weeks	UAM: SUA significantly ↑ during RM intervention MS: •BMI, body fat mass and BW significantly ↓ in the WG group compared to baseline and after washout •creatinine significantly ↑ during RM intervention GB: • *Collinsella aerofaciens* appearing after WG intervention • *Clostridium* sp. ↑ after RM intervention
Low-fat and restricted-calorie diet; Low-carbohydrate and non–restricted-calorie diet Mediterranean and restricted-calorie; [58]	/	235 participants with moderate obesity (ages 40 to 65 years)	24 months	UAM: •SUA ↓ at 6 months and 24 months among all participants •the effect of SUA ↓ in all group was positively correlated with baseline MS: BW, HDL-C, TC: HDL-C, TG, insulin resistance significant improved in all three groups
1.5 L of a mineral water with 2.673 mg HCO_3_/L [59]	The same amount of water with 98 mg HCO_3_/L	34 patients with multiepisodic calcium oxalate urolithiasis (average age of 52.7 years)	Not specified	UAM in the intervention group: •UUA supersaturation, significant ↓ •pH -value in the intervention group, significant ↑ (*p* < 0.001)
Total energy value: 40% from carbohydrates, 30% from proteins and 30% from lipids, <300 mg/day of fatty acids and cholesterol (RESMENA group) [60]	Total energy value: 55% from carbohydrates, 15% from proteins and other treatments were the same as the intervention group	41 women and 52 men with metabolic syndrome (ages 40 to 65 years)	6 months	UAM: SUA significantly ↑ in the control group vs. baseline MS: •waist circumference, BMI, BW, waist: hip ratio, android fat mass and alanine aminotransferase and aspartate aminotransferase significantly ↓ in RESMENA group vs. baseline • glucose and aminotransferase significantly ↑ in the control group •LDL-C and HDL-C significantly ↑ in treatment groups vs. baseline
Isocaloric diets: 30% of energy from animal (AP) or plant (PP) protein [61]	/	44 type 2 diabetes patients (ages 18 to 80 years)	6 weeks	UAM: SUA ↓ in both groups MS: •M-value of insulin sensitivity significantly ↑ in the AP group vs. baseline •TC, LDL-C, HDL-C ↓ in both groups •fasting nonesterified fatty acids significant ↓ in the PP group vs. baseline •CRP significantly ↓ in the AP group
DASH diet [62]	The typical American diet.	103 prehypertensive or hypertensive adults (ages ≥ 22 years)	90 days	UAM: •SUA ↓ at 30 and 90 days in the DASH group •SUA ↓ at 30 and 90 days in the DASH group when participants with baseline SUA ≥6 mg/dL
Soy protein trial: soy protein group (soy protein and isoflavones); isoflavone group (milk protein and isoflavone); Soy flour trial: whole soy group (soy flour); daidzein group (low-fat milk powder and daidzein) [63]	Soy protein trial: milk protein Soy flour trial: low-fat milk powder	450 postmenopausal women with either prediabetes or prehypertension (ages 48 to 65 years)	6 months	UAM: •SUA significantly ↓ in the soy flour and soy protein groups (SCF group) compared with the isoflavone and daidzein groups and the milk placebo groups (MP group) •UA net decrease and UA% decrease between the SCF group and the MP group
Drinking filtered soup (250 g of fresh *Phaseolus Vulgaris* + 1000 mL water) at least an hour before breakfast every other day [64]	/	5168 subjects (ages ≥ 40 years)	6 weeks	UAM: SUA significantly ↓ in the intervention group
Rice bran oil plus a standard diet (RBO) [65]	Sunflower oil plus a standard diet (SO)	40 patients with severe CAD undergoing angioplasty (ages 30 to 70 years)	8 weeks	UAM: SUA ↓ in the RBO group MS: TG, BG, TC, LDL and TNF-α ↓ in the RBO group
The powders of lotus root and cucumber (first, they were squeezed into juices, and then freeze-dried under vacuum) in warm water [66]	/	25 men and 9 women (ages > 60 years)	30 days	UAM: •PUA ↓ in both of lotus root group and cucumber group MS: •plasma glutathione peroxidase ↑ in both of lotus root group and cucumber group •blood mononuclear cell DNA damage ↓ in the lotus root group

Note: data are from clinical trials that have been included in PubMed since 2012. A direct search was used to search for the following terms: “diet and uric acid” or “diet and gout” or “food and uric acid” or “food and gout”. A total of 462 articles were obtained. After following these exclusion criteria—repetitive articles, acute trials, dietary supplements, combination of drugs and food, questionnaire survey, exercise interference, and mismatched intervention subjects—a total of 32 articles showed the effect of diet on uric acid and other indicators of metabolic syndrome. ↑—increase; ↓—decrease; AA—amino acid; AUC—lower area under the curves; BG—blood glucose; BW—body weight; CRP—*C*-reactive protein; DASH—Dietary Approaches to Stop Hypertension; Ereq—energy requirement; HDL—high density lipoprotein; ICAM-1—intercellular cell adhesion molecule-1; LDL—low density lipoprotein; MDA—malondialdehyde; MS—indicators of metabolic syndrome; OR—odds ratio; PUA—plasma uric acid; SUA—serum uric acid; TBARS—thiobarbituric acid reactive substance. TC—total cholesterol; TG—triglyceride; UAM—indicators of uric acid metabolism; UUA—urine uric acid.

**Table 2 nutrients-14-03525-t002:** Combination of nutrition modification and drug supplementation.

Medicine	Dietary Intervention	Participants	Time	Major Findings
Lesinurad [126]	High-fat and high-calorie meal	16 healthy men (ages 18 to 55 years)	6 days	•C_max_ ↓ vs. the fasted phase •serum urate-lowering effect and renal clearance ↑ vs. the fasted phase •absorption was slightly delayed vs. fasted phase
Lesinurad [127]	Moderate-fat diet	16 nonobese men (ages 18 to 55 years)	10 days	•T_max_ 4 h delay •C_max_ ↓ in the fed state vs. the fasted phase
Colchicine [128]	Seville orange juice or grapefruit juice	44 nonobese adults (ages 18 to 45 years)	4 days	•C_max_ and AUC ↓ in the seville orange juice group vs. the nonjuice group •T_max_ occurred 1 h delay compared with in the seville orange juice group vs. the nonjuice group
Febuxostat [129]	High-fat breakfast	68 healthy adults (ages 18 to 55 years)	Not specified	•C_max_ and AUC ↓ under feeding conditions vs. fasting conditions •SUA concentrations ↓ after treatment with febuxostat (80 mg)
Etoricoxib [130]	High-fat meal	12 healthy adults (ages 50 to 64 years)	10 days	•the rate of absorption ↓ in the fed phase vs. the fasted phase •T_max_ occurred with an approximately 2 h delay in the fed phase vs. the fasted phase
Allopurinol/oxipurinol [125]	High-protein or low-protein diet	6 healthy adults (ages 20 to 30 years)	28 days	•plasma AUC significantly ↑ in the high-protein diet group •renal clearance significantly ↓ in the high-protein diet group
Allopurinol [131]	Low-purine diet	60 hypertensive patients with high SUA levels (average age of 54.4 years)	36 weeks	•SUA significantly ↓ in the intervention groups •6 months after the intervention, SUA shows an elevation tendency in the low-purine diet + medication group and medication-only group •6 months after the intervention, SUA shows a continuous drop in the low-purine diet group

Note: Data are from clinical trials that have been included in PubMed. A direct search was used to search for the following terms: “diet” or “food” with “drug” or “medicine” or “treatment” with “gout” or “hyperuricemia”; “diet” or “food” with commonly used clinical anti-gout drugs including “zurampic (lesinurad)”, “colchicine”, “febuxostat”, “allopurinol”, “probenecid”, “aspirin”, “pegloticase”, “benzbromarone” and “etoricoxib”. A total of 751 articles were obtained. After following these exclusion criteria—repetitive articles, dietary supplements, drug interactions, formulation improvement, exercise interference, nonmarket food and mismatched intervention subjects—a total of 7 articles showed the effect of the combination of diet and medication in the treatment of gout. ↑—increase; ↓—decrease; AUC—area under curve; C_max_—maximal plasma concentration; SUA—serum uric acid; T_max_—time to reach C_max_.

## Data Availability

Data available in a publicly accessible repository.

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
