# Peer review of "Gout and Diet: A Comprehensive Review of Mechanisms and Management"

_nutrients, 2022, doi:10.3390/nu14173525_

Round 1

Reviewer 1 Report

Review diet & gout

Thank for the opportunity to review this manuscript. The authors compile current literature and knowledge about the diets impact on gout regarding both mechanism and management in a comprehensive manner. The manuscript is very well-written and describes the pathways in which the diet affect gout in an apprehensible form. In general, good reviews or overviews written on this topic is scarce, and this review will fill a much needed void. I especially like that the authors discusses the possible interaction between diet and drugs - an issue often forgotten. I only have a few remarks on the manuscript:

Minor issues

  1. Although the review is comprehensive, I would not classify it as systematic. To be systematic it should have strict inclusion/exclusion criteria described as well as definitions on how literature was searched. Consequently, I strongly suggest that the title should be altered to just ”review” or perhaps ”comprehensive review”

  2. Page 12 line 337. I would not classify ref 63 as a clinical trial, authors should rephrase this statement. Perhaps cohort, or observational study, may be more appropriate

  3. Page 4, line 147-148. I don’t understand the sentence ”…diet is mainly designed to provide calories or energy expenditure…”, should it state ”…for energy expenditure”?
  4. Sentences start with abbreviations. Although it is a matter of traditions whether to start a sentence with an abbreviation or not, it may in this manuscript be helpful for the reader to have them written out more than just the first time, since several of the abbreviations are not well-known for the general reader.

Author Response

Dear Reviewer,

I would like to express our sincere appreciations on your insightful comments on our manuscript entitled “Gout and diet: a systematic review of mechanisms and management” (Manuscript ID: nutrients-1882602). We have carefully taken your comments into account and provided response to each of the points. Please see the attachment for the point-to-point response.

Reviewer 2 Report

Comments to the Authors: Manuscript ID: nutrients-1882602
Title: Gout and diet: a systemic review of mechanisms and management

The authors presented the thorough review on the role of dietary and nutritional factors in the development of gout. The authors also proposed dietary modification strategies for gout management.

The topic is interesting and actual. The presented research is comprehensive and precise. The review presents the current knowledge and valuable suggestions.

Author Response

Dear Reviewer,

Thank you very much for the time and effort spent in reviewing our manuscript entitled “Gout and diet: a systematic review of mechanisms and management” (Manuscript ID: nutrients-1882602). Thank you again for your approval of our manuscript!